

# Health status of the red-billed tropicbird (*Phaethon aethereus*) determined by hematology, biochemistry, blood gases, and physical examination

Alice Skehel[1,2], Catalina Ulloa[2], Diane Deresienski[3,4], Cristina Regalado[2], Juan Pablo Muñoz-Pérez[1,2], Juan Augusto Garcia[5], Britta Denise Hardesty[6,7], Ronald K. Passingham[3], Jason Steve Castañeda[5], Gregory A. Lewbart[2,3] and Carlos A. Valle[2]

[1] Faculty of Science and Engineering, University of the Sunshine Coast, Sippy Downs, Queensland, Australia

[2] Colegio de Ciencias Biológicas y Ambientales COCIBA and Galápagos Science Center GSC, Universidad San Francisco de Quito, Quito, Ecuador

[3] Clinical Sciences, College of Veterinary Medicine, North Carolina State University, Raleigh, NC, United States of America

[4] Colegio de Ciencias de la Salud, Medicina Veterinaria, Campus Cumbayá, Universidad San Francisco de Quito, Quito, Quito, Ecuador

[5] Terrestrial Ecology, Parque Nacional Galápagos Directorate, Puerto Baquerizo Moreno, Galápagos, Ecuador

[6] CSIRO Environment, Canbera, Australia

[7] Center for Marine Sociology, University of Tasmania, Hobart, Tasmania, Australia

## ABSTRACT

The red-billed tropicbird, *Phaethon aethereus*, is a species of seabird native to the Galápagos archipelago, and widely distributed across the neotropics. General health, blood chemistry, and haematology parameters have not been published for this species. Blood analyses were performed on samples drawn from 51 clinically healthy red-billed tropicbirds captured from their burrows at Islote Pitt on San Cristóbal Island in July, 2016 (21) and Daphne Major Island in June, 2017 (30). In the field, a point of care blood analyser (iSTAT) was used to obtain results for $HCO_{3-}$, pH, $pCO_2$, $pO_2$, $TCO_2$, iCa, Na, K, Cl, Hb, HCT, anion gap, creatinine, glucose and urea nitrogen. Additionally, a portable Lactate Plus[TM] analyser was used to measure blood lactate, and blood smears were also created *in situ*. The blood slides were used to estimate leukocyte counts and 100-cell differentials. Alongside these biochemistry and haematology parameters, average heart rate, respiratory rate, body temperature and scaled mass index (calculated from weight and a body measurement) were compared to determine the standard measurements for a healthy individual. The baseline data, and reference intervals reported in this paper are essential to detecting changes in the health of red-billed tropicbirds in the future.

Corresponding author
Gregory A. Lewbart,
greg_lewbart@ncsu.edu

## INTRODUCTION

The red-billed tropicbird (*Phaethon aethereus*) was traditionally considered a member of the Pelecaniformes, in the family Phaethontidae (*Livezey & Zusi, 2007*; *Bourdon, Amaghzaz & Bouya, 2008*). However, this placement has been revised with the rise of whole-genome analysis and tropicbirds are now placed in their own order Phaethontiformes (*Jarvis et al., 2014*). This long-lived pelagic seabird is found across the neotropics, tropical Atlantic, eastern Pacific and the Indian Ocean (*Vilina et al., 1994*; *Javed, Khan & Shah, 2008*; *Nunes, Mancini & Bugoni, 2017*). The subspecies *P.a.mesonauta* (*Orta, 1992*; *Nelson, 2005*) breeds year-round in the Galápagos (*Snow, 1965*). This region holds one of the largest colonies of this species, consisting of several thousand pairs on the Galápagos islands and mainland Ecuador (*Del Hoyo, Elliott & Sargatal, 1992*; *Orta et al., 2019*). Other extant members of this family include the white-tailed tropicbird and the red-tailed tropicbird.

To date, there has been a lack of information available on the health and population trends for red-billed tropicbirds and related species. Among the Phaethontidae, only the red-tailed tropicbird (*Phaethon rubricauda*) of Christmas Island, had leukocyte profiles reported (*Dehnhard, Quillfeldt & Hennicke, 2011*; *Dehnhard & Hennicke, 2013*). This study did not report on biochemistry or blood gases, making our investigation the first of its kind for this family and the focal species.

The Galápagos Archipelagos are located 972 km west of mainland Ecuador. They are a sanctuary for breeding Pacific seabirds and are home to many endemic species. The islands are impacted by climate change, introduced pathogens, parasites and invasive species (*Wikelski et al., 2004*; *Edgar et al., 2008*; *Riofrío-Lazo & Páez-Rosas, 2015*; *Dueñas, Jiménez-Uzcátegui & Bosker, 2021*). Due to the many human-driven changes to the Galápagos, it is important to define 'clinically normal' animals, so in the future, we can identify changes that occur in a species' health status (*Valle et al., 2018*). In this study, we provide a detailed account of 51 individuals anatomical, physiological, and blood-specific values that are widely used to evaluate the health of wild seabirds (*Work, 1996*; *Padilla et al., 2003*; *Padilla et al., 2006*; *Ratliff et al., 2017*; *Lewbart et al., 2017*; *Valle et al., 2018*; *Valle et al., 2020a*; *Valle et al., 2020b*). This work is also relevant because the global population of red-billed tropicbirds are thought to be decreasing although the species is classified as least concern (*BirdLife International, 2019*).

Ground-nesting birds, especially those on islands, are at risk from terrestrial predators, particularly introduced non-native species (*Stratton, Nolte & Payseur, 2021*). In the Galápagos, for example, introduced predators such as pigs, rats, goats and cats threaten breeding birds (*Cruz & Cruz, 1987*; *Key, Wilson & Conner, 1994*; *Riofrío-Lazo & Páez-Rosas, 2015*; *Debrot et al., 2014*). Endemic island species are also susceptible to diseases (*Wikelski et al., 2004*; *Deem et al., 2011*); in the Galápagos, some have been infected by introduced pathogens (*Levin et al., 2012*; *Bastien et al., 2014*; *Tompkins et al., 2017*). Having baseline values from healthy animals provides a foundation for comparison when diseased individuals are identified.

There has been no published work on the health of red-billed tropicbirds, but publications have reported leucocyte profiles and heterophil to lymphocyte ratios for

*P. rubricauda* (*Work, 1996*; *Dehnhard, Quillfeldt & Hennicke, 2011*; *Dehnhard & Hennicke, 2013*). Literature concerning red-tailed tropicbirds on Christmas Island demonstrated leucocyte profiles are not influenced by sex or breeding stage (*Dehnhard & Hennicke, 2013*). They also suggest that chicks of this species invest heavily in innate immunity as they displayed significantly higher heterophil/lymphocyte ratios than adults (*Dehnhard, Quillfeldt & Hennicke, 2011*).

The aim of this study is to present baseline health results for the Galápagos Island red-billed tropicbirds and to be able to identify changes in their health in future.

## MATERIAL AND METHODS

This study was performed as part of a population health assessment authorized by the Galápagos National Park Service (permit No. PC-57-16 and No. PC-59-17 to C.A. Valle) for ethics, animal handling and permissions to work within the Galapagos Reserve (*Valle et al., 2018*; *Valle et al., 2020a*; *Valle et al., 2020b*; *Tucker-Retter et al., 2021*). The study was additionally approved by the Universidad San Francisco de Quito ethics and animal handling protocol. Handling and sampling procedures followed standard vertebrate protocols and veterinary practices. Data were collected in 2016 at Punta Pitt, San Cristóbal Island (0°41′59″S; 89°15′09″W) and in 2017 on Daphne Major Island (0°25′22.14″S; 90°22′16.17″W) from two separate multi-species seabird colonies.

In June 2016 and June 2017, 21 and 30 adult red-billed tropicbirds, respectively, were captured from their nesting holes, one at a time. At time of capture, the nesting status was recorded, where possible, alongside the "age of chick" and "presence of eggs" or "other adult". The adults were identified as distinct from the chicks by their plumage and bill colour. The birds were restrained carefully by one handler, holding the bill closed, whilst supporting the weight of the body and restraining the wings. The behaviour of the bird was monitored closely during handling to ensure its safety. Within 10 min of capture, heart rate and respiratory rate were recorded, as well as a blood sample quickly and safely drawn. This short turnaround time was to reduce the potential effects of handling on blood chemistry values. The birds were then weighed, using a Pesola (Pesola, Prazisionswaagen AG, Schindellegi, Switzerland) measured for standard body measurements, and had their cloacal temperature recorded using a digital thermometer. All birds were deemed clinically healthy based on their behaviour, response to handling, and physical examination by a veterinarian. Sex could not be distinguished since red-billed tropicbirds are monomorphic (*Nunes et al., 2013*).

Brachial vein puncture was performed using a heparinized 25-gauge needle and 1.0 ml syringe to collect up to 1.0 ml of blood. To obtain the blood gas, electrolyte, and biochemistry results, several drops of blood were used for instant analysis in the field, with a Portable iSTAT Clinical Analyser (Heska Corporation, Loveland, Colorado, USA) (*Harter et al., 2015*). Different iSTAT cartridges were used for different years, CG8+ *vs* CHEM8+, which meant the recorded health parameters differed slightly. In 2017, pH, $HCO_3^-$, $pCO_2$ and $pO_2$ were not obtained. Another drop of blood was then used to obtain lactate levels through a portable lactate analyser. To obtain total solids, a field centrifuge

was used to spin haematocrit capillary tubes for 3 min, and the plasma in the capillary tubes was dropped onto a handheld refractometer. Calculated iSTAT haematocrits were obtained for all individuals. In addition, blood smears were also created in the field, and were later fixed and stained with Diff Quick (Jorgenson Laboratories, Loveland, CO, USA) approximately 2 weeks after sampling.

The stained blood smears were used for the 100-cell white blood cell (WBC/leukocyte) differentials and for WBC estimates (*Work, 1996*). The WBC differentials were obtained at a 100 × objective lens and completed by differentiating 100 white blood cells. This was recorded as a percentage and these were used to calculate absolute values using the total leukocyte count (estimated WBC count). Leukocytes were counted in 10 fields at a 40 × objective. These were averaged and multiplied by 2000 to obtain a leukocyte count per μL (*Newman, Piatt & White, 1997*). Blood smears from 2016 ($N = 21$) and 2017 ($N = 30$) were analysed by different observers, however, to minimise observer differences, the subsequent observer was trained and supervised by the observer from 2016.

Scaled mass index (*Peig & Green, 2009*; *Peig & Green, 2010*) was calculated using the R package (standardized) major axis estimation and testing routines 'smatr' 3.4–8 (*Warton, Wright & Wang, 2011*) was used for computing the scaling exponent using standardized major axis (SMA) regression. Summary statistics were completed on the measured parameters. Normality was tested using Shapiro–Wilk test and visualised through quantile–quantile plots. Despite 28 out of the 34 variables of the variables being non-normal, the sample size is sufficiently large to invoke the central limit theorem meaning linear regressions could be used to examine possible relationships between SMI (scaled mass index) and log transformed blood biochemistry and haematology differentials (*Sainani, 2012*). Sampling time was tested against the variables using Kendall's rank correlation, a test chosen to handle non-normal data with outliers, reliably. Significant results were then examined with linear regressions on log transformed data. A standard $\alpha$ level of $P < 0.05$ was used for all statistical tests with R statistical software, version 3.6.2 (*R Core Team, 2019*) and figures were made using *ggplot2* (*Wickham, 2009*). Reference intervals were calculated and are reported for health parameters (*Walton, 2001*). They were calculated using Reference Value Advisor freeware (*Geffré et al., 2011*) with a sample number <40 and >40, using robust methods, Box–Cox transforming data and removing outliers. Outliers were identified through Tukey's testing and histograms, and emphasis was put upon retaining rather than deleting data (*Horn & Pesce, 2003*). Alongside this, the median, minimum and maximum values have been made available as Supplementary Information to support in clinical decisions (*Friedrichs et al., 2011*; Klassen, 1999).

## RESULTS

In 2016 and 2017 a total of 51 adults of *P. aethereus* from two different colonies were assessed; the blood chemistry, morphology and haematology results alongside reference intervals are presented in Tables 1, 2 and 3.

The dominant leukocytes for *P. aethereus* at Islote Pitt and Daphne Major were lymphocytes ($\mu = 34.2\%$, absolute $\mu = 995 \times 10^3/\mu$L), eosinophils ($\mu = 30\%$; absolute $\mu$

**Table 1  Physiological measurements and blood sample time at Punta Pitt.** Summarising the mean, standard deviation, range and reference intervals for morphological and physiological measurements, alongside sample time, for 21 red billed tropicbirds (*Phaethon aethereus*) at Punta Pitt, San Cristobal, and 30 on Daphne Major.

| Parameter | Red-billed tropicbird values | | | | | Reference values | |
|---|---|---|---|---|---|---|---|
| | $\mu$ | $\sigma$ | Median | Range | $n$ | Reference intervals | Missing (%) |
| Wing length (cm) | 31.2 | 2.21 | 31.0 | 28.7–39.0 | 51 | 28.7–33.1 | 6 |
| Tarsus length (cm) | 3.20 | 0.501 | 3.14 | 2.20–4.96 | 51 | 2.25–3.96 | 2 |
| Scaled Mass Index | 504 | 186 | 444 | 265–1590 | 51 | 282–681 | 2 |
| Weight (kg) | 0.59 | 0.07 | 0.60 | 0.30–0.73 | 51 | 0.52–0.70 | 8 |
| Respiratory rate (beats/min$^{-1}$) | 33.7 | 9.93 | 32.0 | 16.0–60.0 | 50 | 16.0–57.8 | 4 |
| Heart rate (beats/min) | 237 | 60.9 | 244 | 144–500 | 51 | 145.8–458 | 2 |
| Body temperature (°C) | 39.6 | 1.0 | 39.6 | 37.7–41.6 | 51 | 37.8–41.6 | 0 |
| Sample time (min:second) | 7:98 | 6:32 | 6:00 | 2:00–38:00 | 46 | N/A | N/A |
| Handling time (min:second) | 15:00 | 6:55 | 13:00 | 5:00–40:00 | 48 | N/A | N/A |

**Table 2  Mean blood values at Punta Pitt.** Summarising the mean, standard deviation, range and reference intervals (including the interval and the percentage of values that were removed as outliers) for blood gas, biochemical and haematology values for 21 red billed tropicbirds (*Phaethon aethereus*) at Punta Pitt, San Cristóbal and 30 on Daphne Major.

| Parameter | Red-billed tropicbird values | | | | | Reference values | |
|---|---|---|---|---|---|---|---|
| | $\mu$ | $\sigma$ | Median | Range | $n$ | Reference intervals | Missing % |
| HCT % | 36.4 | 5.62 | 37.0 | 18.0–49.0 | 50 | 25.9–47.9 | 4 |
| Total Solids (g/L) | 58 | 18.2 | 53.5 | 32–108 | 38 | 3.19–10.42 | 8 |
| Haemoglobin (g/L) | 124 | 191 | 126 | 61–167 | 50 | 99–150 | 8 |
| pH (37 °C) | 7.41 | 0.092 | 7.38 | 7.26–7.69 | 21 | 7.19–7.59 | 0 |
| pH (Auto-corrected) | 7.37 | 0.089 | 7.35 | 7.24–7.67 | 21 | 7.24–7.49 | 0 |
| HCO3- (mmol/L) | 17.4 | 3.41 | 16.8 | 11.9–25.4 | 21 | 11.7–26.23 | 0 |
| pCO2 (mmHg) | 27.7 | 6.12 | 27.8 | 13.3–37.3 | 21 | 12.3–39.0 | 0 |
| pO2 (mmHg) | 58.1 | 11.9 | 58.0 | 40.0–87.0 | 21 | 35.9–86.7 | 0 |
| Na (mmol/L) | 140 | 5.21 | 142 | 116–148 | 50 | 134–148 | 6 |
| K (mmol/L) | 3.48 | 0.699 | 3.40 | 2.40–6.60 | 50 | 2.44–4.08 | 6 |
| iCa (mmol/L) | 0.951 | 0.258 | 1.00 | 0.00–1.30 | 50 | 0.62–1.30 | 8 |
| Glucose (mmol/L) | 16.5 | 3.19 | 16.9 | 5.94–23.8 | 50 | 9.5–23.3 | 2 |
| Lactate (mmol/L) | 3.78 | 1.40 | 3.60 | 1.20–6.60 | 51 | 1.26–6.57 | 0 |

$= 938 \times 10^3/\mu L$) and heterophils ($\mu = 34.1\%$, absolute $\mu = 908 \times 10^3/\mu L$) (Table 3). No basophils were observed in *P. aethereus*, and monocytes percentages were low.

From the statistical analysis, sampling time was shown to demonstrate significantly strong positive effects on $pO^2$ ($P < 0.05$, tau $= 0.389$) and strong negative impacts on $pCO^2$ ($P < 0.05$, tau $= -0.367$) and $HCO^{3-}$ ($P < 0.05$, tau $= -0.361$) as seen in Fig. 1. A simple linear regression was calculated to predict whether the calculated scaled mass index (SMI) affected the variables and was graphically examined. Outliers were removed and SMI demonstrated no effects on the variables.

Heterophils had a colourless cytoplasm with a high density of red to brown large fusiform granules and a segmented lobed nucleus. Eosinophils had similar size to heterophils, with

**Table 3 White blood cell count values for 21 red billed tropicbirds (*Phaethon aethereus*) at Punta Pitt, San Cristobal, and 30 of the same on Daphne Major.** Summarising the mean, standard deviation, range and reference range for morphological and physiological measurements, alongside sample time, for 21 red billed tropicbirds (*Phaethon aethereus*) at Punta Pitt, San Cristobal, and 30 on Daphne Major.

| Parameter | Red-billed tropicbird values | | | | | Reference values | |
|---|---|---|---|---|---|---|---|
| | $\mu$ | $\sigma$ | Median | Range | $n$ | Reference intervals | Missing % |
| Heterophil % | 34.1 | 12.4 | 33.0 | 8.5–64.5 | 50 | 9.0–56.4 | 2 |
| Monocytes % | 1.66 | 1.32 | 1.50 | 0–4.5 | 50 | 0.0–4.5 | 0 |
| Eosinophils % | 30.0 | 16.8 | 28.0 | 2–68.0 | 50 | 3.1–66.5 | 0 |
| Lymphocytes % | 34.2 | 16.8 | 28.0 | 2.0–68.0 | 50 | 21.7–50.3 | 8 |
| Basophils % | 0 | 0 | 0 | 0–0 | 50 | 0–0 | 0 |
| Estimated WBC $\times 10^3/\mu$L | 2890 | 1380 | 3000 | 820–9910 | 50 | 830–4380 | 2 |
| Absolute Values | | | | | | | |
| Heterophil (ABS) $\times 10^3/\mu$L | 908 | 411 | 825 | 323–2460 | 50 | 323–2460 | 2 |
| Monocytes (ABS) $\times 10^3/\mu$L | 46.8 | 41.7 | 44.3 | 0–198 | 50 | 0–129 | 2 |
| Eosinophils (ABS) $\times 10^3/\mu$L | 938 | 841 | 684 | 64.8–4510 | 50 | 67.5–2160 | 8 |
| Lymphocytes (ABS) $\times 10^3/\mu$L | 995 | 610 | 862 | 246–3910 | 50 | 246–3910 | 4 |
| Basophils (ABS) $\times 10^3/\mu$L | 0 | 0 | 0 | 0–0 | 50 | 0–0 | 0 |

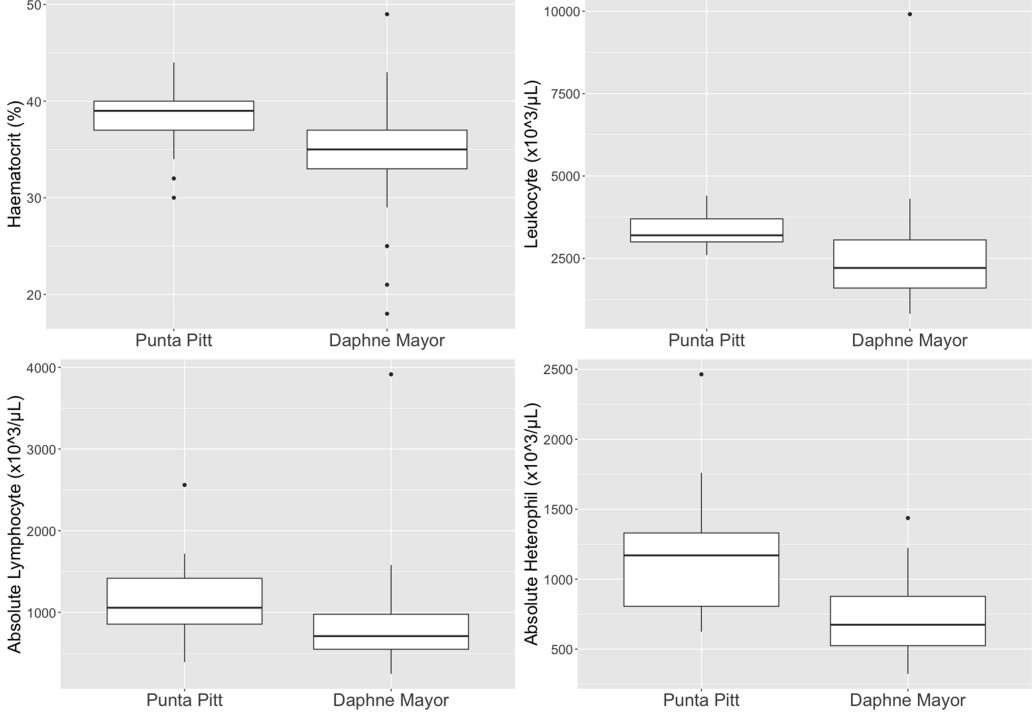

**Figure 1 Blood gases and sampling time correlations.** Linear regression plots demonstrating significant correlations between sampling time and blood gases for red-billed tropicbirds (*Phaethon aethereus*) sampled in Punta Pitt (2016) and Daphne Major (2017).

amorphous to large granules and numerous small clear vacuoles, in accordance with *P. rubricauda* (*Work, 1996*) for birds sampled from both populations.

No hemoparasites were observed for red-billed tropicbirds at either site.

## DISCUSSION

This is the first haematological profile of *P. aethereus*. This study reports an in-depth array of morphometric, vital, blood biochemistry, and haematological parameters from red-billed tropicbirds. No other studies present health assessments for this species of tropicbird and the results from this study will be a reliable baseline for future health assessments. Reference intervals for the health parameters are presented in this paper to ensure that there is information to compare to if any further studies follow the population health of these tropicbird populations. Healthy adult individuals were directly selected for this baseline health assessment and no sex differential studies could be made with sufficient confidence since these birds are monomorphic (*Nunes et al., 2013*). Additionally, reproductive state was not always evident. Preanalytical procedures for reference intervals usually including standardising the provisioning of the individuals to be either fasted, or non-fasted, however this is not possible in wild populations. All sample capture, handling, sample collection and processing were standardised. Outliers identified by the Tukey's test were examined and only accepted after examining all the data from the individual at once, as well as investigating a histogram of the transformed data. Identified outliers were removed after this thorough investigating and before calculating the reference intervals. Smaller sizes make this more difficult but the robust method is utilised as it puts less emphasis on the extreme values (*Horn & Pesce, 2003*).

Leukocyte estimates for *P. aethereus* ($\mu = 2890 \times 10^3/\mu L, \sigma = 1380 \times 10^3/\mu L$) were low relative to other seabird species studied in the Galápagos during 2016 and 2017 at Daphne Major and Punta Pitt, including great frigatebirds ($\mu = 3700.37 \times 10^3/\mu L, \sigma = 693.37 \times 10^3/\mu L$), Galápagos shearwaters ($\mu = 3283 \times 10^3/\mu L, \sigma = 6720 \times 10^3/\mu L$), swallow-tailed gulls ($\mu = 3928 \times 10^3/\mu L, \sigma = 1098 \times 10^3/\mu L$), and Nazca boobies ($\mu = 17553 \times 10^3/\mu L, \sigma = 3333 \times 10^3/\mu L$ (*Valle et al., 2018*; *Valle et al., 2020a*; *Valle et al., 2020b*; *Tucker-Retter et al., 2021*)).

The leukocyte differentials demonstrated no basophils and low to no monocyte counts. These white blood cells are both uncommon in avian peripheral blood (*Fudge, 1994*). The leukocyte profiles showed almost equal percentage counts of heterophils, lymphocytes and eosinophils. This percent of eosinophils is high relative to the eosinophil percentage of other populations of *P. rubricauda* (*Dehnhard, Quillfeldt & Hennicke, 2011*; *Dehnhard & Hennicke, 2013*; *Luna et al., 2020*). Eosinophils can vary greatly between avian species therefore the differences noted here between the *P. aethereus* and *P. rubricauda* is not unusual (*Fudge, 1994*) and the reported percentages presented in this paper are the known baseline for this population. When comparing other studies' leukocyte differentials, as a percentage of 100 counted cells, Phaethontidae species have variable percentage eosinophil counts. The population of red-billed tropicbirds in Galápagos had a relatively high eosinophil percentage count ($\mu = 30\%$), as well as the adult red-tailed tropicbirds on Rapa Nui, which had $\mu = 36.69\%$ (*Luna et al., 2020*). In contrast, the red-tailed tropicbirds on Christmas Island, had very low eosinophil counts ($\mu = 0–1\%$) (*Dehnhard, Quillfeldt*

*& Hennicke, 2011*; *Dehnhard & Hennicke, 2013*). Total leukocyte estimates and manual differential counts have high coefficients of variation relative to automated nucleated cell counts, which is a limitation of this study but was the only practical approach under field conditions.

Stress is known to have adverse effects on wild animals (*Sagar et al., 2019*; *McMahon, Hoff & Burton, 2005*). Although this has never been studied in Phaethontiformes, handling time was kept short to ensure minimal stress to the individual (*Sorenson et al., 2016*).

When sampling time increased, $pCO_2$ and $HCO_3$ decreased, while $pO_2$ increased.

This is likely due to the birds' increased respiratory rate due to stress during capture and handling. The changes in the blood chemistry can be explained through Le Châtelier's principle (*Hopkins, Sanvictores & Sharma, 2022*), with deep or rapid breathing drawing away more $CO^2$ and bringing in $pO^2$ more rapidly, increasing the $pO^2$ levels and lowering the $pCO^2$. This disturbs the dynamic equilibrium of the primary pH buffer system and the position of the equilibrium will shift to reestablish an equilibrium, which is seen by the decrease in $HCO^{3-}$. If a change in pH was seen this could lead to acute metabolic alkalosis (*Fudge, 1994*).

Seabird species in the Galápagos Archipelago are at risk since they are vulnerable to human population growth on the Islands, tourism, and movement between the archipelago and Ecuador (*Walsh & Mena, 2016*). Other pressures include introduced rodents predating ground-nesting species in the Galápagos and although *P. aethereus* were not referenced in this study (*Harper & Carrion, 2021*), other populations of this species face serious population decline due to chick and egg predation from the rat species found in Galápagos, including both invasive house (*Rattus rattus*) and brown rats (*Rattus norvegicus*) (*Sarmento et al., 2014*).

Few health and physiology studies have been completed on red-billed tropicbirds due to the animals' remote breeding habitat on steep rocky cliffs on remote islands. This study creates a foundation of veterinary knowledge for this species and contributes to general knowledge of seabird health. This information will be an essential comparison as increasing human presence and pressure on this vulnerable ecosystem places small seabird rookeries at risk. Continuing to monitor and understand species that live on the edge will be vital to their conservation and the conservation of other coastal species.

## ACKNOWLEDGEMENTS

We thank Galo Quezada, Maryuri Yepez, Diego Quiroga, Carlos Mena, Stephen Walsh, Philip Page, Tillie Laws and Michael Levy for their support and assistance with this project.

### Funding

This research was conducted with the support of the Heska Corporation, the Galápagos Academic Institute for the Arts and Sciences (GAIAS)-Universidad San Francisco de Quito (USFQ) and the Galápagos Science Center-USFQ/University of North Carolina-Chapel

Hill. The funders had no role in study design, data collection and analysis, decision to publish, or preparation of the manuscript.

### Grant Disclosures

The following grant information was disclosed by the authors:

Heska Corporation, the Galápagos Academic Institute for the Arts and Sciences (GAIAS)-Universidad San Francisco de Quito (USFQ).

Galápagos Science Center-USFQ/University of North Carolina-Chapel Hill.

### Competing Interests

The authors declare there are no competing interests.

### Author Contributions

- Alice Skehel conceived and designed the experiments, performed the experiments, analyzed the data, prepared figures and/or tables, authored or reviewed drafts of the article, and approved the final draft.
- Catalina Ulloa conceived and designed the experiments, performed the experiments, analyzed the data, prepared figures and/or tables, authored or reviewed drafts of the article, and approved the final draft.
- Diane Deresienski conceived and designed the experiments, performed the experiments, authored or reviewed drafts of the article, and approved the final draft.
- Cristina Regalado performed the experiments, authored or reviewed drafts of the article, and approved the final draft.
- Juan Pablo Muñoz-Pérez performed the experiments, authored or reviewed drafts of the article, and approved the final draft.
- Juan Augusto Garcia performed the experiments, authored or reviewed drafts of the article, and approved the final draft.
- Britta Denise Hardesty performed the experiments, authored or reviewed drafts of the article, and approved the final draft.
- Ronald K. Passingham performed the experiments, authored or reviewed drafts of the article, and approved the final draft.
- Jason Steve Castañeda performed the experiments, authored or reviewed drafts of the article, and approved the final draft.
- Gregory A. Lewbart conceived and designed the experiments, performed the experiments, analyzed the data, authored or reviewed drafts of the article, and approved the final draft.
- Carlos A. Valle conceived and designed the experiments, performed the experiments, analyzed the data, prepared figures and/or tables, authored or reviewed drafts of the article, and approved the final draft.

### Animal Ethics

The following information was supplied relating to ethical approvals (*i.e.*, approving body and any reference numbers):

This study approved by the Universidad San Francisco de Quito ethics and animal handling protocol. All handling and sampling procedures were consistent with standard vertebrate protocols and veterinary practices.
## Field Study Permissions

The following information was supplied relating to field study approvals (*i.e.*, approving body and any reference numbers):

This study was performed as part of a population health assessment authorized by the Galapagos National Park Service (permit no. PC-57-16 and no. PC-59-17 to C.A.V.).

## Data Availability

The raw data is available in the Supplemental File.

## Supplemental Information

Supplemental information for this article can be found online at http://dx.doi.org/10.7717/peerj.15713#supplemental-information.

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
