# Peer review of "Health status of the red-billed tropicbird (Phaethon aethereus) determined by hematology, biochemistry, blood gases, and physical examination"

_PeerJ, doi:10.7717/peerj.15713_

## Round 0.1 · original submission · Major Revisions

Please note that both reviewers have identified concerns regarding measurements and statistical analyses that preclude several of the conclusions from this study. If you are able to address these concerns, a revision can be submitted.

Reviewer 1 ·

Basic reporting

The article is generally well written with only a few spelling mistakes or ambiguous phrasing. The reference and background information provided are appropriate except for 1-2 instances (see additional comments).
The figures and tables are good, although some units or explanations should be added (see comments).
I would say that the main weakness of the paper is the methodology, for the complete blood count component especially, uncertainty about the health status (eg: possible eosinophilia?), and possibly some sampling or machine error at one sampling site (HCT and total protein differences).

Experimental design

I think the goal is well stated: reporting normals, for future reference.
However, there is some question about the actual health status of these birds and the validity of some of the results both due to the technique and potential for spurious results in one colony. Also, if the goal is to use these data for future comparison to establish a health status then the study should aim to provide normal reference intervals (see guidelines for doing this by the American Society of Veterinary Clinical Pathologists (ASVCP)), not just descriptive statistics.

Validity of the findings

Please see further comments about this in the additional comments below.

Additional comments

Please find below some suggested changes and comments for your consideration:

line 37 - Change leucocytes to leukocytes

l84 - Change gender to sex, how were the birds restrained (any sedation?), Is there any way to guess the age?

The fact that a CBC was done (estimate leukocyte count and 100-cell differential) is not currently listed in the abstract. Please add this information.

l93 - Hematocrit was not available for some samples in 2016....was it available for any of them? If not, change wording to say it wasn't available for any of them, if so, please mention how many were available.

l94 - Please explain how you obtained the estimated WBC count from the smears and what you mean by: you modified from a 40x to a 100x objective. Do you mean you counted the leukocytes in 10 fields at 100x? What formula did you use to calculate the WBC count then? Counting at 100x instead of 40x means you count fewer cells in total, which means your counts are likely less reliable than the 40x counts already are. Also, the Newman paper reports correcting the PCV/HCT. Did you do this? if so, which number did you use as the PCV/HCT you considered normal? and did you use the manually obtained hematocrit in these birds? What did you do for those for which this was not available?

l100 - T-test for comparison: comparing between what and what? Survey site? If so, this should be brought up as a goal of the study before the results section.

l102 - Summary statistics are ok but if used to make medical decisions as stated as part of the objectives then reference intervals should be calculated instead.

l114 - If stats show a difference between the groups is the difference to a degree that is clinically significant? (you can use the total allowable error data in the ASVCP guidelines to get a general idea of the type of difference that is considered significant). Is one colony unhealthy then or were there issues with the samples or machines to explain the differences? If these findings are real then perhaps stratification of the data is warranted. Or are there other things that differed between the colonies (eg: nutrition, age)?

l119-22 - The wording is a bit confusing here. The data is similar or different to what? previous papers? or do you mean between colonies? Note that the Dehnard papers does not provide absolute counts for the leukocytes either (aka: number of cell per liter or microliter depending on if it's scientific or empirical units) so I would not compare this data with those papers unless you mention that you are talking specifically about the %, which is not as useful clinically as the #/ul or #/L.

l139 - Typo in the word 'heterophils'

l144 - Please add a paragraph with discussion of the absolute values for the leukocytes . Also, absolute values of leukocytes are rarely normally distributed (they are usually skewed, at least the absolute values) so discussing the means in this part of the text is likely not an appropriate approach.

l152 - Please re-insert reference for this information

l166 - Please discuss why you think that these differences are observed, especially the differences in the HCT and total protein seen in figure 1. Do you think that these are real findings? If not (and I really don't think that it is), what do you think could have interfered with their measurement/calculations? Which data is wrong? Should it be discarded?

l168 - Again here, I would call eosinophilia based on absolute counts, not just the %. The absolute counts may also be increased, however, in which case the discussion still stands. Except for the fact that maybe these are not healthy individuals after all.

l171 - Were there ectoparasites only at Punta Pitt or in both site?

Figure 1 & 3 - Include units for the WBC count

Figure 4 - I don't think that this figure adds much to the paper given that the data is presented in figure 1 already and we can see that iSTAT calculated HCT is very low for DAPH but that manual HCT is closer between the colonies, which is likely why there is some difference seen in this figure.

Figure 5 - Is it possible to get a higher resolution, white balanced image for these again? I am not certain that I can see the difference. For example B looks to me like it is a heterophil. However, this may be an issue with the image.

Table 2 & 3 - Could you please define in the legend what the numbers in the 'Value' column are? (mean +/- SD (range?)). In the last two rows, are those min:seconds or minutes with decimals? what does 8:64 mean? 8 minutes and 38 seconds? Are the values reported in the value column appropriate for the distribution of the data (aka: are these all normally distributed?). For tables 3 & 4, please include the units for the WBC, and also report out the absolute value for each type of leukocyte as, clinically, the % are of little value and it is the absolute counts that matter most. On that note, could you clarify if the statistical comparison was done on the % or the absolute values for these?

·

Basic reporting

Clear use of English.
There are some issues in figures and tables. There is one missing table (I think it’s table 2), affecting the understanding of the entire section. Other comments are detailed on the review file.

Experimental design

Here we have the major issue of the manuscript: there are some statistical errors, detailed on the review file.

Validity of the findings

Some statistical adjustments are required, but findings are valuable.

Additional comments

Dear Author,

Congratulations on your research, but there are some important points of improvements, as it follows:
1. Your most important issues:

- Material and Methods: the presence of just one non-normal variable is enough to use only non-parametric tests. You cannot mix parametric and non-parametric tests in the same research. With that said, it will be necessary to redo all statistical analyses, using just non-parametric tests. I cannot review the results and discussion related to the statistical analysis until the correction of this issue.
- Results:
o There is one missing table (I think it’s table 2), affecting the understanding of the entire section.
o Line 119: you cannot talk about significant difference using just a table. Statistical analyses are required to affirm that.
o Lines 124 to 138 need to be rewritten after the statistical analysis correction. As well as lines 163 – 166 in the Discussion section.

2. Other important issues:
- Abstract: you should add the results, at least the most important ones.
- Material and methods:
o Do you have the permission number of the Universidad San Francisco de Quito ethics and animal handling? You should write it on the text, like the permission of the Galapagos National Park Service that is on line 71.
o Line 82: you should describe the instruments used to weight and to record the body temperature of the birds.
o Line 100: you should first describe the normality test used and after that, you write about other tests. Summary statistics also should come before the tests in the text.
o Line 101: the variable is non-normal; the test is non-parametric. With that said, you should replace “non-parametric” with “non-normal variables”.
- Results: lines 120 to 122 are difficult to understand. Which blood chemistry parameters are different between populations? And which one had high eosinophil counts? They seem so similar to me, as well as leucity profiles. Can you rewrite it?
o Lines 150 – 152: Do you have any supposition about the reason why the eosinophil counts are higher than other studies? If so, can you add it on the discussion?
o Line 154: you should write “BUN” unabbreviated at this first time.
- Figure 1: In subtitles, you say that the “boxplots demonstrate statistically significant differences between…” You cannot demonstrate statistically significant differences using a graphic. Statistical analyses are required. Also, you have to add the acronym meanings in subtitles.
- Figure 4: “comparison of red-billed tropicbirds between two breeding colonies” – it is not what the graphic is demonstrating. It is demonstrating comparison between the two methods.
- Table 1: there is no need to explain what is sample time in the subtitle. Keep it short. You can explain it on the text.

3. The least important points:
- Introduction: line 62 – isn´t it “published work” instead of “work published”?
- Material and methods: lines 86 and 87 – if possible, you should write down the needles and the syringes brands.

4. Strong points:
- Very important research. It is valuable to have available data from the species, especially in free-living birds.
- I admire your care in keeping short the handling and sampling time.
- You have excellent recommendations on future studies along the text. Maybe you could reunite all this recommendations at the end of the text.
- Valuable micrographs illustration.
- Excellent use of English language.

Congratulations!

---

## Round 0.2 · Major Revisions

While the revisions have improved the manuscript, there remain substantial concerns with the statistical analysis and multiple occasions of inaccurate use of units and grammar. Both reviewers have highlighted these concerns in an constructive manner, and the findings from your study will be clearer and more meaningful once these concerns have been addressed.

Reviewer 1 ·

Basic reporting

Please see comments below. The main issue currently is the units. Some less important comments on grammatical errors and suggested wording changes to wording for clarity are included below. Also, since many hematologic parameters are not normally distributed the statistics used in the last table are likely not appropriate. A link is provided below for guidance on how to do this correctly.

Experimental design

No comment

Validity of the findings

No comment

Additional comments

Page 1: line 15: Change blood slides to blood smears
Page 4: line 102: Is "were" supposed to be "where"?
Page 5: line 150: change "total protein" to "total solids"
Line 147 and 151, p7 line 282 etc…: change “manual hematocrit levels” to “packed cell volumes” or PCV for future mentions of this.
Line 160-1: Change sentence to: this was recorded as a percentage and these were used to calculate the absolute values using the estimated total WBC count.
Page 6: line 191: … and multiplied by 2000… add “to obtain a leukocyte count per μL”
P7 line 284 change "to" to "on"
P8 line 344-347 - I think that the units must be incorrect here. I could see a mean heterophil count being 11.7 x 10^9/L or 11.7 x 10^3/μL being correct but 11.7x10^4/L does not make sense for a healthy bird population. Same for the other leukocytes including the absolute values for eosinophils further down. Also, I would recommend choosing either 10^3/μL (empirical units, used in the USA) or 10^9/L (scientific units, used in the rest of the world) and sticking with these units throughout for consistency.

All mention of hematology parameters should come with appropriate units, regardless of if other papers have managed to get published without them or not. A value without a unit is meaningless. You can chose to report SI units throughout the paper or empirical units or both but I recommend being consistent. For example: hemoglobin is reported as either g/L (SI units) or g/dL (empirical units). Calcium is in mmol/L (SI units) or mg/dL (empirical). Find out which units your machines reported for reach parameter and amend the tables and figured accordingly. For the CBC, consider finding a hematology textbook, avian medicine textbook or veterinary publication with reference intervals for any species (ideally a recent publication, perhaps in the Veterinary Clinical Pathology journal) to see what units are typically used but also to get a sense of what the absolute values for leukocytes and biochemistry parameters typically are in other birds. They are not all that variable across species so if yours are not close I would double check the units you are using.

Figure 1 and 2: Add units in the legend and in the graphs after checking that these are indeed the units used by the machine. iCa is ionized calcium (as opposed to total calcium which are often given by other analyzers). In figure 2, the total est WBC is likely in #/uL here based on the numbers you have represented, not 10^9/L. Convert all leukocyte counts to either # x10^3/uL or # x 10^9/L and stick with this throughout the text and figures.

Table 2: The manually obtained HCT the way you’ve done it is called a packed cell volume (PCV), hematocrit is calculated. Therefore you could change HCTM to PCV and HCTI to just HCT

Table 3: Thank you for adding this table, I think this will be of great value once it is fixed. The units for the absolute values are wrong. Please correct these as mentioned above. Also please also use ASVCP guidelines for guidance on how to do the stats to report this and include a sentence in the text mentioning that the guidelines were followed, or if not followed: why. https://cdn.ymaws.com/www.asvcp.org/resource/resmgr/QALS/Other_Publications/RI_Guidelines_For_ASVCP_webs.pdf (see section 11.4 with instructions on reporting these for sample sizes between 20 and 40 individuals).

·

Basic reporting

No comment

Experimental design

No comment

Validity of the findings

No comment

Additional comments

Dear Author,
You have corrected the most important issues of your manuscript, but there are some minor adjustments required, as it follows:

- Materials and methods:
- Line 139: replace “non-parametric” with “non-normal”.
- Lines 140 – 144: As your results are non-normal, you must apply Spearman’s correlation test instead of Pearson’s correlation. Please correct it and see if the results remain the same. Also check if linear regression applied is suitable for non-normal results.
- Results and Discussion:
- Lines 174 – 175: please, rewrite it after correcting the statistical analysis (Spearman´s correlation). You also have to present the result of the test (rho), in addition to p-value, describing the strength of the correlation and if it is positive or negative.
- Every significant result should be discussed. With that said, there are some points that could be well discussed:
o Differences between handling time among sites. Why did it happen? (Results presented on lines 175-177);
o Differences noted on lines 210-212. Can you assume why did it happen? Are there ecological differences among sites?
- Figure 1 (subtitles): Keep it simple. The statistical analyses are on the text. Here you are just showing the levels, not the statistical differences. Suggestion: "Standard boxplots demonstrating levels of haematocrit (HCT), haemoglobin, Calcium (ICa) (check unit) and glucose (check unit) for red-billed tropicbirds (Phaethon aethereus) in Punta Pitt (PITT) (2016) and Daphne Major (DAPH) (2017), Galapagos archipelago, Ecuador."
- Figure 2 (subtitles): Keep it simple. The statistical analyses are on text. Here you are just showing the levels, not the statistical differences. Suggestion: "Standard boxplots demonstrating levels of heterophis and lymphocitytes, white blood cell count (check unit) for red-billed tropicbirds (Phaethon aethereus) in Punta Pitt (PITT) (2016) and Daphne Major (DAPH) (2017), Galapagos archipelago, Ecuador."
- Table 1 (subtitles): remove the sentence “which is time taken to obtain the blood sample from capture”. It is already explained on the text.

Congratulations!
Best regards.

---

## Round 0.3 · Major Revisions

The revised manuscript continues to have unclear and/or inaccurate wording and descriptions. While obtaining the data undoubtedly took much effort, the results derived need to be interpreted more cautiously with understanding that methodological inconsistencies rather than biological features may account for much of the observed differences.

1. For example, line 134: "...heart rate and respiratory rate were recorded, as well as a blood sample quickly and safely drawn." should be "a blood sample WAS quickly drawn". Weren't all blood samples drawn "safely"? What constituted unsafe?

2. "This short turnaround time was to reduce the potential effects of handling on blood chemistry values." That effect may manifest later than in 10 minutes - consistent time frame is more important than short time frame.

3. "Pesola" and "Covidien" are trade names. Source should be indicated.

4. L166: "The rest of the blood was stored on ice in sterile plastic vials within 10 minutes of sample collection for haematology and future analyses." Stored for how long and what analysis was done later on whole blood?

5. L173: "These were averaged and multiplied by 2000 to obtain a Leukocyte count per µL" An estimate would be the best that can be derived by this method (not a count).

6. L174: Considering that the numbers derived in this study are based on very variable manual methods, having the same individual (rather than 2 different ones) perform 100 cell differential counts on ~50 slides would slightly increase robustness of data.

7. L348: What do you mean with 'robust' methods?

8. L359: Lymphocytes are not granulocytes.

9. L453: That is really unclear. 29% less of what?

10. L455: All gases should have sub- rather than superscripts. And why would samples collected in one year differ in PO2/PCO2 (PCO2 and HCO3 are essentially the same) from another year? That is likely methodological error rather than a biological difference.

11. L469: The data presented here would at best be a 'guide' but certainly not a 'reliable baseline".

12. L572: That is not correct. See point 5 above.

13. L576: "...this thorough investigating and before calculating the reference intervals." This sentence does not make sense.

L676: "Sample time (the time from capturing the individual until taking the blood sample) was also kept short to minimise changes in the blood chemistry due to stress. " That would only apply to effects of catecholamines. Stress mediated by glucocorticoids may manifest over 24 or more hours.

L678: "When sampling time increased some correlation could be seen relating to respiratory chemistry and the primary pH buffer system, with pCO2 and HCO3- decreasing and pO2 increasing as time until sampling increased." This sentence does not make sense.

Fig. 2: The images are not of equal magnification and have green background. Fig. 2C has higher magnification than A, B and D. The originals need to be white-balanced. Same color problem in Fig. 3.

Table 2: What does "*" refer to? The units for total solids are wrong - should be 58 g/L and not 5.8 g/L. Where are 'reference values' derived from?

---

## Round 0.4 · Minor Revisions

I have amended a few wordings in the attached file. Please review. The blood smear images are of poor color and resolution, and while it is possible to 'recognize' different cell types, details cannot be discerned. If you have the smears at hand, it would be much preferred if the images could be retaken with white balance and high resolution.

---

## Round 0.5 · accepted · Accept

Thanks for the final response.